# Enhancing Orthotic Treatment for Scoliosis: Development of Body Pressure Mapping Knitwear with Integrated FBG Sensors

**DOI:** 10.3390/s25051284

**Published:** 2025-02-20

**Authors:** Ka-Po Lee, Zhijun Wang, Lin Zheng, Ruixin Liang, Queenie Fok, Chao Lu, Linyue Lu, Jason Pui-Yin Cheung, Kit-Lun Yick, Joanne Yip

**Affiliations:** 1School of Fashion and Textiles, The Hong Kong Polytechnic University, Hong Kong 999077, China; kpllee@polyu.edu.hk (K.-P.L.); lin.zheng@polyu.edu.hk (L.Z.); ruixin.liang@polyu.edu.hk (R.L.); lh-queenie.fok@connect.polyu.hk (Q.F.); kit-lun.yick@polyu.edu.hk (K.-L.Y.); 2Foshan Fifth People’s Hospital (Foshan Rehabilitation Hospital), Foshan 528211, China; fswykf@126.com; 3Photonics Research Institute, Department of Electrical and Electronic Engineering, The Hong Kong Polytechnic University, Hong Kong 999077, China; chao.lu@polyu.edu.hk (C.L.); lycat.lu@polyu.edu.hk (L.L.); 4Department of Orthopaedics and Traumatology, The University of Hong Kong, Hong Kong 999077, China; cheungjp@hku.hk

**Keywords:** smart textiles, optical fiber sensor, inlay, knitted undergarments, fiber Bragg grating, health monitoring, scoliosis, pressure monitoring, force monitoring

## Abstract

Bracing is a widely used conservative treatment for adolescent idiopathic scoliosis (AIS) patients, yet there is no consensus on the optimal amount of force applied. Although a number of different sensors have been developed to continuously monitor the applied pressure and force, they have several limitations, including inadequate overall force distribution and displacement. They also cause discomfort with limited wearability. In this study, body pressure mapping knitwear (BPMK) integrated with fourteen silicone-embedded fiber Bragg grating (FBG) sensors is developed to monitor immediate and overall changes in force during the bracing treatment. A wear trial of the BPMK is conducted by using a validated soft AIS mannequin, and prediction equations have been formulated for the FBG sensors at individual locations. The findings indicate that the measured forces are in good agreement with those obtained from clinical studies, with peak forces around the padding regions reaching approximately 2N. This was further validated by using finite element (FE) models. When comparing X-ray images, the estimated differences in Cobb angles were found to be 0.6° for the thoracic region and 2.1° for the lumbar region. This model is expected to provide valuable insights into optimal force application, thus minimizing the risk of injury and enhancing bracing compliance and efficacy. Ultimately, this innovative approach provides clinicians with data-driven insights for safer and more effective bracing applications, thus improving the quality of life of AIS patients.

## 1. Introduction

Adolescent idiopathic scoliosis (AIS) is a three-dimensional (3D) spinal deformity that affects 1.7% of adolescents who are between 10 and 18 years old worldwide [1]. This condition often requires long-term orthotic treatment to manage its progression [2,3]. AIS is characterized by the lateral curvature of the spine of more than 10 degrees accompanied by vertebral rotation, which leads to physical concerns if left untreated, such as asymmetrical shoulder blades or waistline and functional impairments like reduced lung capacity [4,5]. The current golden standard for managing scoliosis involves the use of rigid orthoses to apply corrective forces to the torso and spine. However, the effectiveness of these orthotic devices is greatly dependent on maintaining the prescribed levels and duration of pressure on the torso of the AIS patient throughout his/her daily activities [6,7,8].

Despite the widespread use of orthoses, there is no consensus on the optimal force applied with bracing, so the applied amount of force is left to the professional discretion of the orthotist [9,10]. Along with the challenges posed by the different spinal curvatures of the patients and their different levels of tolerance, a major reason for inconsistent standardized force measurements is the significant differences among the sensors used. For example, Lou et al. [11] used a commercial force transducer—the FS01 force sensors (Honeywell International Inc., Charlotte, NC, USA)—with the Boston Brace and reported an average exerted force of 6.78 N. However, Dehzangi et al. [12] used the FSS1500nsb force sensor (Honeywell International Inc.) with the Boston Brace and found that it is capable of exerting forces that range from 300 N to 500 N onto the torso. Although the sensors were positioned at different locations on different subjects, the considerable variations in the applied force may challenge orthotists who rely on force measurements to adjust the tightness of the brace. Aside from point sensors, Hudák et al. [13] used Pressurex-micro, a pressure-indicating film, to measure the dynamic and overall pressure distribution, and the measured forces ranged from 0.013 to 0.137 MPa. The color of the film changes based on the applied pressure, with a higher pressure resulting in a more intense dark color. However, monitoring that provides immediate results is not available because the data storage system requires time to process, which makes it difficult to adjust the pressure during brace fitting. Lee [14] found that a customized sensing system can be pre-embedded into the braces but the process is costly and time-consuming.

Integrating fiber Bragg grating (FBG) sensors into orthoses has been promising to continuously measure the forces exerted onto the body of patients, thus allowing for immediate adjustments to maintain optimal corrective pressure [9]. Moreover, Rothmaier et al. [15] indicated that FBG sensors are particularly well-suited for integration with textiles, owing to their cost effectiveness, adaptability, lightweight nature, durability, and non-toxic composition. Furthermore, the potential applications of these sensors are significantly increased by their high strain sensitivity and resistance to electromagnetic interference [16,17,18]. Additionally, optical fiber sensors address the limitations associated with conventional electronic sensors. For example, during magnetic resonance imaging (MRI), optical fiber strain sensors can be employed to monitor real-time respiratory activity [19], a task that is not feasible with electronic sensors that contain metal components. To address these challenges, this research paper explores the development of a novel smart knitwear solution with integrated FBG sensors, which is designed to enhance the effectiveness and acceptability of scoliosis treatment.

Recruiting human subjects to evaluate the response of the spine to bracing force is a conventional yet time-consuming approach that poses significant risks due to the unknown effects of the different magnitudes of force exerted on the body [20]. Furthermore, such studies often necessitate a number of X-ray images, which expose the patients to radiation which has harmful health consequences. Consequently, researchers have been exploring alternative methodologies, such as using finite element analysis (FEA), which is a safer and more efficient means of studying the response of the spine without subjecting patients to radiation or other high-risk procedures. FEAs are a numerical method used to solve engineering and physical problems without extensive experimentation [21,22]. Their ability to save time and reduce risks has made them invaluable for evaluating orthopedic devices and optimizing biomechanical designs [23]. Previous studies have demonstrated the potential of FEAs in predicting the efficacy of bracing and determining the optimal force parameters [21,22,24]. However, the effectiveness of FEAs is contingent on the use of reliable data, namely the loading conditions and mechanical properties; otherwise, their feasibility may be compromised. For instance, Fok [21] and Chan [22] used loading data collected from Pliance^®^ sensors in FEAs to predict the efficacy of textile braces. Notably, Fok [21] further utilized an FEA to identify the optimal corrective levels. However, both studies are limited by their use of only two areas of loading, which may not adequately represent the pressure exerted by the brace. Therefore, it is essential to investigate how overall pressure distribution influences spinal corrective outcomes.

This paper investigates the response of Cobb angles and spinal curvature to bracing and builds on the FE model in Fok [21]. The study encompasses the design and development of body pressure mapping knitwear (BPMK) with fourteen FBG sensors, which enable the measurement of both the overall and immediate forces exerted by braces. Additionally, a brace wear trial is conducted by using a validated soft mannequin that simulates AIS [20]. During this trial, the forces exerted by the brace are recorded by the BPMK. The collected data are subsequently used to refine the FE model. This enhancement allows the biomechanical model to accurately simulate the forces applied by the brace and analyze the response of the spine to the exerted pressure, thereby validating the BPMK. The validation of this model yields detailed insights into the optimal amount of force that can be applied through bracing, thereby minimizing the risk of injury and enhancing bracing compliance and efficacy. Furthermore, the BPMK can provide real-time feedback on treatment effectiveness, thus facilitating timely adjustments.

## 2. Methods

### 2.1. FE Model and AIS Soft Mannequin

The optimal amount of corrective pressure or force applied to evaluate the immediate effects of bracing has been traditionally determined through conventional X-ray imaging sessions. However, this practice raises significant ethical concerns due to the unnecessary exposure of patients to radiation. To circumvent this issue, this study applies the validated FE model in Fok [21]. This approach not only removes the ethical concerns associated with radiation exposure but also enhances the efficiency of the assessment process.

The FE model was developed based on a 12-year-old female patient with AIS. Her Cobb angles were 25.3° in the thoracic (T5 to T11) and 21.8° in the lumbar (T11 to L3) regions, with a Risser grade of 2 (Table 1). She was prescribed a functional intimate apparel (FIA) otherwise known as the anisotropic textile brace (ATB) in Fok [21] for 2 h of wear, which is a common practice in clinical settings to determine the initial correction of the spine. This innovative brace is constructed from breathable textiles, elastic bands and straps, semi-rigid pads, and 3D-printed artificial bones. According to the clinical data, the pressure exerted by the brace can be adjusted by changing the tension of the elastic bands and straps, which can be up to 24.4 kPa (2.44N) [25]. Following the bracing intervention, her Cobb angles decreased to 18.7° and 18.8° in the thoracic and lumbar regions, respectively.

Given the difficulties in recruiting another subject with the same Cobb angles and surface contours, a validated soft AIS mannequin [20], developed based on the measurements of the subject [21], was used in lieu. The soft mannequin was designed to accurately measure the interface pressure between the brace and skin of the patient, and evaluate the efficacy of bracing for patients with AIS. The assembled mannequin is a close replicate of the AIS patient in Fok [21], with similar torso contours, spinal curvature and Cobb angle based on clinical X-ray images of the subject and shows the effects of wearing the brace. This approach ensures consistency and reliability in simulating the biomechanical conditions relevant to the study. Additionally, measurements performed on the soft mannequin would not be subjected to interferences such as body movement and respiratory activity, thereby reducing the duration of measurements. Using the soft mannequin not only enhances the accuracy of the collected data but also confirms that the observed corrective effects on the spine are due to compressional forces, rather than axial forces resulting from bodily movements.

### 2.2. Force Measurement on AIS Soft Mannequin

#### 2.2.1. Design of FBG Sensor Series

The loading conditions in the FEA in Fok [21] were derived from clinical pressure measurements taken from the subject with the use of the Pliance^®^-xf-16 system (Novel, Munich, Germany) equipped with six single pressure sensors. In this study, however, the BPMK uses fourteen FBG sensors to measure the force exerted by the FIA.

Previous experimental findings by Lee et al. [9] indicated that FBG sensors embedded in silicone exhibit more stability and linearity than the Pliance^®^ pressure sensors when tested on soft surfaces. However, all four FBG sensors in their study were placed at the back of the undergarment, which may not be able to comprehensively provide overall force distribution. To enhance the data collection process, fourteen FBG sensors were designed to be placed onto the BPMK. However, to address the installation challenges, costs, and time associated with fabricating fourteen FBG sensors into a single optical fiber, the sensors were distributed across four optical fibers. This strategy is meant to enhance the efficiency of the fabrication process. The intervals between each FBG sensor were carefully measured prior to their inscription on the optical fibers (Figure 1).

#### 2.2.2. Integration of FBG Sensors onto BPMK

Subsequently, the sensors were embedded into a silicone membrane as in Lee et al. [9] and inlaid into a pure PIMA cotton knitted undergarment in the warp direction (Figure 2a). A silicone membrane of 1.5 mm in thickness was fabricated by using Dragon Skin^®^ 20 silicone (Figure 2b) after which a curvilinear groove was made onto the membrane. Subsequently, an FBG sensor was inserted into the BPMK and embedded in the groove of the silicone membrane (Figure 2c). Figure 2d shows the cross-section of an embedded FBG sensor within the knitted structure of the BPMK. This embedding process not only enhances the elasticity and sensitivity of the sensors but also minimizes the likelihood that they would shift during use. This procedure was repeated fourteen times until all of the FBG sensors were integrated into the silicone membranes.

As illustrated in Figure 3, the fourteen FBG sensors are strategically positioned at key areas that correspond to the brace pads, including the following: (1) ribs, (2) anterior superior iliac spine at the pelvis, (3) underarm, (4) waist, (5) iliac crest at the pelvis, (6) thoracic muscles, and (7) lumbar muscles. To enhance sensor identification, the FBG sensors are arranged in pairs and labeled as “R” and “L”, which denote the right and left sensors, respectively. The specifications for these FBG sensors are detailed in Table 2.

To obtain the force–strain relationship for interpreting the Bragg wavelength shifts, a simulated wear trial was conducted, as shown in Figure 4. The experimental setup utilizes validated artificial tissues [22] and a force gauge (model: JSV-H1000, Japan Instrumentation System, ALGOL Instrument Co., Ltd., Taoyuan City, Taiwan) to replicate human tissues and apply force with the brace. Artificial tissues of 20 mm in thickness were created by using Ecoflex^®^ 0010 silicone rubber and polyurethane foam, and used to simulate the mechanical properties of the skin and muscles. This synthetic construct has been validated to exhibit properties analogous to those of human skin, thus ensuring its effectiveness for experimental applications [22]. The embedded FBG sensors in the BPMK were positioned between the force gauge and artificial tissues to facilitate accurate force measurements, and the Bragg wavelength shifts were recorded using the sm130 Optical Sensing interrogator (Micron Optics Inc., Atlanta, GA, USA). The force gauge applied 0.5 N to the FBG sensors every 5 s until reaching 10 N. This process resulted in fourteen equations to calculate the applied force (x) at specific locations by inputting the Bragg wavelength shift (y) into the corresponding equation.

To perform the force measurements, the BPMK was placed on the mannequin, which was then fitted with the FIA. To replicate the previous clinical trial conducted on human subjects, silicone pads were strategically positioned within the FIA at the right thoracic and left pelvic regions. The straps were then tightened to the point where the spine began to align and straighten. The test was conducted for approximately one minute, which is sufficient time for the signals to stabilize.

#### 2.2.3. Data Analysis

The data collected from the wear trials were used to assess the linearity (R^2^) between the Bragg wavelength shifts and a range of standardized forces. After the analysis was conducted, fourteen equations were derived, and, therefore, the corresponding force was calculated based on the observed Bragg wavelength shifts of the FBG sensors. The initial Bragg wavelengths ranged from 1550.113 nm to 1555.799 nm for FBG Series 1, 1549.739 nm to 1555.996 nm for FBG Series 2, 1550.127 nm to 1553.980 nm for FBG Series 3, and 1549.997 nm to 1553.995 nm for FBG Series 4 (Table 2).

### 2.3. FE Simulation

#### 2.3.1. Geometric Models Construction

The FE model was developed based on our previous research, incorporating three primary components, namely the torso, skeletal structure, and FIA (Figure 5) [21]. These components were validated for accuracy. The torso and FIA models were constructed from 3D body scans of the subject by using Geomagic Studio 2012 (64-bit) and SolidWorks 2018 (64-bit). Additionally, the skeletal structure model was developed based on X-ray images, which provides an accurate anatomical representation that is critical for evaluating the biomechanical effects of the brace in a simulated environment.

The estimated difference in Cobb angles is 2.1° for the thoracic region and 0.6° for the lumbar region when comparing the X-ray images to the results of an FEA. The good agreement between the simulated and measured results underscores the reliability of the FEA in accurately simulating spinal curvature [26].

#### 2.3.2. Mechanical Properties and Meshing Definition

The geometric models were implemented in MSC Marc 2020 software (version 2022.2.0, Newport Beach, CA, USA), which has been demonstrated to be effective and accurate for simulating human body behavior [27,28]. Since the FE model in Fok [21] has already been validated, the primary modifications involved re-meshing the model to increase its accuracy, and adjusting the material properties and loading conditions to align with those of the soft mannequin and force values collected from the BPMK, respectively. A summary of these properties is presented in Table 3 [21,29,30,31,32]. Notably, the Young’s modulus for the textile material is obtained from stretch and recovery tests performed by using the Instron 4411 tensile strength tester [26]. The Poisson’s ratio is derived from Zhou et al. [33].

One primary modification involved re-meshing the model to make it more precise and time-saving. Elastic–plastic isotropic material type, 3-node triangle elements and 4-node linear quadratic tetrahedral elements were used, each with three degrees of freedom at each node. After the mesh elements were established, a numerical simulation was conducted to evaluate the process of donning the FIA onto the torso.

#### 2.3.3. Boundary and Loading Conditions

The initial and boundary conditions were then defined, which included the displacement conditions of the FIA and loading conditions on the surface of the torso. These boundary conditions were established to accurately simulate real-life scenarios, by focusing on the spinal effects of using the brace. Consequently, the model included the torso and skeletal structure only, with the assumption that the forces exerted by the FIA do not cause displacement of the head or legs. To facilitate this, the top surface of the T1 vertebra and the bottom surface of the torso were constrained by fixing all degrees of freedom (Figure 6a). Additionally, displacement boundary conditions were used to simulate the wear process (Figure 6b). During this process, corrective force was applied to model the interaction between the functional intimate apparel and the torso.

#### 2.3.4. Validation: Comparing Cobb Angles and Spinal Curvature

To evaluate the spinal and postural changes following bracing treatment, it is essential to compare the Cobb angles and spinal curvatures obtained from X-ray imaging, as well as postural evaluations derived from the 3D body images of the subjects. Therefore, an effective validation approach involves comparing the estimated Cobb angles and spinal curvature obtained from an FEA with actual observations from X-ray images [21]. Cobb angles are typically assessed with the use of 2D anteroposterior radiographs taken in a standing position, which primarily quantifies scoliosis in the frontal plane [34]. The Cobb angles were measured before and after applying the brace by using RadiAnt DICOM Viewer (64-bit). This measurement involves calculating the angle formed by drawing tangents on the upper and lower endplates of the most tilted upper and lower end vertebrae [35] (Figure 7).

To compare the spinal curvatures, the position of each vertebra was obtained by measuring the horizontal distance between the vertebral body and the central sacral vertical line (CSVL) by using SolidWorks 2018. Subsequently, the correlation coefficient between the vertebral positions in the FEA and X-rays was calculated. A higher correlation coefficient that approaches 1 indicates a modeled curvature that is in good agreement with the measured curvature in the X-ray images. This approach ensures the accuracy and credibility of the simulated results, thereby enhancing the robustness of the research findings.

## 3. Results and Discussion

### 3.1. Linearity of FBG Sensors Embedded in Silicone

An FBG is created by periodically modifying the refractive index of the optical fiber core through exposure to spatially modulated laser light [36,37]. It is widely utilized for measuring strain and temperature. However, when paired with the appropriate interface, the shift in the Bragg wavelength can serve as a transducer for various other mechanical parameters, such as pressure and displacement [38]. When broadband light is launched into the fiber with this refractive index modulation, a narrowband component is reflected. The Bragg wavelength (λB), which indicates the resonance condition of the grating, is given by:(1)$$\lambda B={2n}_{eff}\Lambda \;,$$
where λB denotes the Bragg wavelength, $n_{eff}$
*denotes* effective refractive index, and Λ denotes the grating pitch.

When FBG sensors are subjected to external perturbations, a corresponding shift in the Bragg wavelength (ΔλB) can be detected. This shift occurs due to changes in the refractive index ($n_{eff}$) and the grating pitch (Λ). In our experiment, the external perturbation was the force applied by the force gauge, which induced a measurable shift in the Bragg wavelength. For instance, applying a force of 1 N resulted in a shift of 0.02 nm in the Bragg wavelength. By analyzing the linear relationship between them, the corresponding force (x) within the specified range of 10 N can be accurately determined by inputting the change in Bragg wavelength shift (y) into the corresponding equation:2R: y = 0.0134x + 0.0190,(2)1R: y = 0.0078x + 0.0125,(3)1L: y = 0.0093x + 0.0152,(4)2L: y = 0.0086x − 0.0032,(5)7R: y = 0.0183x + 0.0340,(6)6R: y = 0.0169x + 0.0295,(7)6L: y = 0.0118x + 0.0185,(8)7L: y = 0.0133x − 0.0143,(9)3R: y = 0.0113x + 0.0366,(10)4R: y = 0.0066x + 0.0202,(11)5R: y = 0.0053x + 0.0071,(12)3L: y = 0.0198x + 0.0262,(13)4L: y = 0.0175x + 0.0072, (14)5L: y = 0.0144x − 0.0075.(15)
where y denotes Bragg wavelength shift, and x denotes force.

Referring to Figure 8, the results show that the range of the linearity (R^2^) values between the force and Bragg wavelength shifts vary from 0.83 to 0.99. This range indicates a high degree of linearity of the FBG sensors in measuring the forces on a soft surface, which may be attributed to the silicone embedding methods.

Traditionally, silica optical fiber sensors are known for their rigidity and fragility, making bending and compression challenging. However, integrating pre-bent FBG sensors within flexible silicone membranes greatly enhances their flexibility, softness, and compressibility, allowing for broader applications in various settings [39,40].

### 3.2. Force Distribution

During testing on a soft mannequin, the sensing locations were strategically placed at the ribs, underarms, waist, anterior superior iliac spine, and iliac crest of the pelvis. Using the 14 equations, the force exerted by the FIA onto the mannequin was determined, and the corresponding values were calculated (Table 4). The data were subsequently incorporated into the FEA model (Figure 9).

It is important to note that these sensing locations differ from those referenced in the clinical data, which may influence the comparison. However, both datasets indicate higher force values near the apex regions, namely T8-T9, and L1 (Figure 9). Specifically, the clinical data show forces of 2.44 N and 0.96 N at the thoracic and lumbar apex regions [25], while the soft mannequin testing yielded forces of 1.4 N and 2.1 N, respectively. This indicates a consistent trend in force distribution, underscoring the relevance of our findings despite the differences in testing conditions.

Additionally, it is crucial to acknowledge the significant variability observed in the sensitivity of the FBG sensors. The variation in the maximum Bragg wavelength shifts serves as a key indicator of the force sensing range and sensitivity, with a larger wavelength shift signifying higher sensitivity [9]. While most of the FBG sensors showed high sensitivity to force on a soft surface, a subset of sensors exhibited markedly lower sensitivity. For example, the FBG sensor positioned at 3L recorded a maximum Bragg wavelength shift of 0.215 nm under a 10 N applied force, whereas the sensor at 5R exhibited only a 0.055 nm shift. This discrepancy in sensitivity could be attributed to factors related to the fabrication process of inscribing the FBG on the optical fiber or potential variations during the inlaying process. Additionally, differences in fabric structure and thickness of the BPMK may also influence sensor sensitivity.

### 3.3. Evaluation of Spinal Correction Effects

Figure 10a illustrates the displacement of the skeletal model post-analysis. From the back view of the subject, it is evident that the thoracic vertebrae show lateral shifts to the left, while the lumbar vertebrae shift to the right. Notably, the T8 and T9 vertebrae, which constitute the apex of the thoracic curvature, are most substantially displaced. These findings suggest that the spine tends to straighten, which is in agreement with the outcomes observed in [21] (Figure 10b).

Table 5 provides a comparative assessment of the Cobb angles before and after intervention with the FIA. It shows a reduction in Cobb angles for both the thoracic and lumbar regions, with differences of 0.6° and 2.1°, respectively.

A comprehensive visual analysis was then conducted in which the estimated spinal curvatures based on the FEA were graphically compared with the actual curvatures derived from X-ray imaging. This comparative analysis was conducted by quantifying the horizontal displacement between the vertebral bodies and CSVL. As shown in Figure 11a,b, both the modelled and measured in-brace spinal curves exhibit a leftward shift from T7 to T10 and a rightward shift from T12 to L4. This alignment indicates a consistent displacement pattern between them. Furthermore, the correlation coefficient between the vertebral positions was determined to be 0.87, which is a much higher correlation than that of the correlation coefficient of 0.74 in Fok [21]. This higher correlation might be attributed to the more accurate estimation of movement in the thoracic (T5 to T8) and lower lumbar (L1 to L4) regions (Figure 11c).

The Cobb angles and spinal curvatures obtained from the FEA modeling and measured clinical data are in good agreement which serves as evidence that the FEA can accurately represent the biomechanical characteristics of AIS, thereby validating this method. Additionally, this shows the reliability of the BPMK which can effectively capture the mechanical forces that act on the spine and reflect the in-brace results. This good agreement between the modeled and measured results enhances the credibility of using FBG sensors as an effective device for immediate assessment and intervention.

### 3.4. Comparison with Traditional Methods

Traditional methods for measuring pressure and force in orthopedic braces have several significant limitations, including the inability to provide overall pressure distribution, issues with sensor displacement, discomfort during wear, and restricted overall wearability. For instance, Chase et al. [41] and Lou et al. [42] used pressure sensors to assess the cushioning forces of braces in the lateral direction; however, their approach is limited to measuring pressure at a single localized area. Similarly, Ali et al. [43] used sensors to evaluate bracing force. Nevertheless, their methods often rely on block-shaped sensors that can only be attached to strap-based braces. These restrictions limit the ability to capture the overall pressure distribution on the body so that they cannot accurately obtain the pressure variations throughout the brace. Additionally, the rigidity of these sensors inhibits their ability to adapt to the natural movement of the body, which potentially leads to discomfort and inaccurate readings due to sensor displacement. While embedding the sensors within a brace could address this issue, the process often requires the customization of design, which is both time-consuming and resource-intensive. Furthermore, the limited wearability of traditional designs poses a significant challenge. As shown in Périé et al. [44], the integration of 192 thin polymer force sensors still results in bulky and rigid support structures, so that continuous monitoring becomes uncomfortable and impractical for patients.

The BPMK, which incorporates fourteen FBG sensors, effectively addresses the limitations of existing systems by enabling both overall and immediate measurements of force. This capability provides a more comprehensive and accurate understanding of the interaction of the brace with the body. Additionally, the strategic integration of FBG sensors within elastic knitwear enhances both comfort and efficiency, thus allowing the BPMK to accommodate individuals with different body shapes and thereby eliminating the need for customized designs. Furthermore, the flexibility of the FBG sensors mitigates issues related to sensor displacement, as they adapt to the natural movement of the body, thus ensuring accurate measurements even during dynamic activities. This adaptability not only enhances wearability but also promotes continuous use, which results in consistent and reliable data collection processes.

### 3.5. Potential Applications

FBG sensors, known for their high accuracy, flexibility, and compatibility with MRI, enhance functionality across various applications.

In the bracing treatment for AIS, the BPMK is believed to potentially fill a critical knowledge gap: “What is the appropriate pressure or force to apply for optimal compliance and corrective effect?”. The BPMK facilitates continuous monitoring of in-brace force distribution for researchers and clinicians, thereby overcoming the limitations of traditional block-shaped sensors, which often lack sufficient measurement points and adaptability to dynamic movement [9]. Additionally, when supported by a detailed FEA, it enables clinicians to accurately evaluate spinal displacement and rotation, along with their effects on soft tissues and neural structures. It implies this combination contributes to a comprehensive understanding of the relationship between the forces applied by the brace and its corrective influence on spinal deformities. Furthermore, the exceptional MRI compatibility of BPMK enables respiratory monitoring during scans. They can also be integrated into orthopedic cast padding to monitor force or pressure on wounds, which contributes to preventing complications from overly tight-fitting casts and enhancing patient safety and wear comfort during the healing process.

In the long term, the collected data can be utilized to establish guidelines for braces, including optimal pressure levels, recommended padding areas, and appropriate strap tensions, ultimately mitigating the risk of physical injuries. The improved brace accuracy reduces the incidence of excessive brace pressures and inefficient bracing, thereby minimizing the negative impact on patients’ daily activities and self-esteem. Therefore, the brace compliance and overall effectiveness of bracing treatment can be significantly improved. The BPMK is expected to lead to better treatment compliance and efficiency, ultimately empowering patients with a reliable and supportive intervention that fosters their overall well-being and quality of life.

In health-related products like compression socks [45], the integration of FBG sensors enables real-time evaluation of pressure distribution. This ensures uniform compression, thus enhancing circulation and reducing swelling which are key factors in preventing conditions such as deep vein thrombosis and facilitating early detection of diabetic foot ulcers. They can also be used in smart footwear, which analyses gait and pressure distribution to prevent falls among the elderly.

In the sportswear field, integrating FBG sensors into leggings and sports bras gives athletes valuable insights into muscle compression, posture alignment, and impact forces during physical activities [9,27]. These data are crucial for optimizing performance, preventing injuries, and guiding recovery protocols. For the elderly, FBG sensors embedded into mattresses or bed linens enable continuous monitoring of body movement, which can prevent pressure ulcers and enhance safety and comfort in long-term care settings. These sensors can alert caregivers to prolonged pressure on specific areas, which can facilitate timely intervention and improve patient outcomes.

## 4. Limitations of Experiments and Future Works

One significant limitation identified in the experiment is the fragility of the silica optical fiber, which can be easily damaged. Additionally, the extended length of the optical fiber connected to the FBG interrogator may impact the wearability of the BPMK. This necessitates heightened caution during their application on soft mannequins. Addressing this issue is critical for the successful implementation of the BPMK in clinical settings. A recommended alternative is the use of polymer optical fibers, which have shown enhanced elasticity and flexibility. Additionally, the development of a miniature interrogator would further improve the portability and user-friendliness of the BPMK. Furthermore, the methodology and placement of silicone embedding significantly influence the force–strain relationship, given that FBG sensors exclusively measure strain. Future research should focus on investigating the impact of silicone on the FBG response. A comprehensive understanding of this interaction may enable the use of silicone coatings for precise adjustments, thereby obviating the need to remove the coating from the FBG sensor during inscription. While the BPMK is designed to accommodate a range of body shapes, individuals with very small or large torsos may not receive an optimal fit. To enhance usability and fit, future designs could incorporate front openings, such as button closures, to facilitate ease of adjustment. We believe that using BPMK in human trials is feasible, and this paper serves as an initial step toward that objective. Mitigating these limitations is critical for the successful implementation of testing on human subjects, which will be undertaken following the necessary improvements.

Another limitation of this study is the assumption of the elastic material’s properties, which may not fully represent real-world behavior. Future work will focus on developing FE models that incorporate elastic–plastic properties and detailed geometric representations, such as specific organs, to improve realism and applicability.

Monitoring the amount of applied force is crucial for tracking the progression of spinal deformities and informing treatment plans, which ultimately improve patient outcomes. The combination of FBG technology, MRI, and FEA represents a significant step forward in managing spinal conditions and provides a valuable approach to understanding and addressing the complexities of AIS. In contrast to X-ray assessments, MRI does not present radiographic limitations and is capable of providing high-resolution images almost immediately, which elucidate the effects of the exerted forces on the spinal cord. Therefore, MRI is a valuable method that can be used with the metal-free BPMK to evaluate spinal response to applied forces. Consequently, Cobb angles can be measured in real-time across different magnitudes of force, thus facilitating accurate monitoring of spinal curvature and alignment. This capability is instrumental in determining the optimal amount of applied force prior to prescribing a brace, thereby enhancing treatment efficacy over the length of use. Future research could continue to investigate how these technologies can work together to improve monitoring and clinical decision-making in scoliosis treatment.

## 5. Conclusions

The BPMK effectively measures force distribution, particularly when evaluating the bracing force on soft surfaces. The high linearity of FBG sensors embedded in silicone, reflected in R^2^ values that range from 0.83 to 0.99, confirms their feasibility in detecting these forces. Furthermore, its reliability and accuracy have been validated through an FEA, which provides a comprehensive understanding of the amount of force exerted by the brace. The data collected of the force exerted onto a soft mannequin are incorporated into the FE model, and produce results that are in close agreement with the measured data, including significant reductions in the Cobb angles before and after intervention with the FIA. Compared to traditional methods, the BPMK offers superior overall and immediate force measurements, enhanced wear comfort, and adaptability to body movement; therefore, it is a reliable tool for the continuous monitoring of intervention with AIS braces. The flexibility and MRI compatibility of FBG sensors broaden their applicability across various health-related domains, including orthopedic treatment and sports performance. Overall, the BPMK represents a substantial advancement in biomechanical monitoring, providing critical insights for clinicians and researchers and ultimately improving patient outcomes in spinal health and beyond.

## Figures and Tables

**Figure 1 sensors-25-01284-f001:**
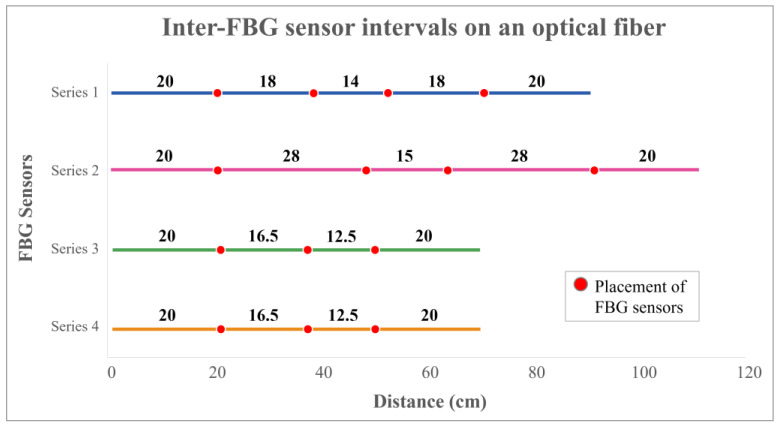
Inter-FBG sensor intervals on four series of optical fibers.

**Figure 2 sensors-25-01284-f002:**
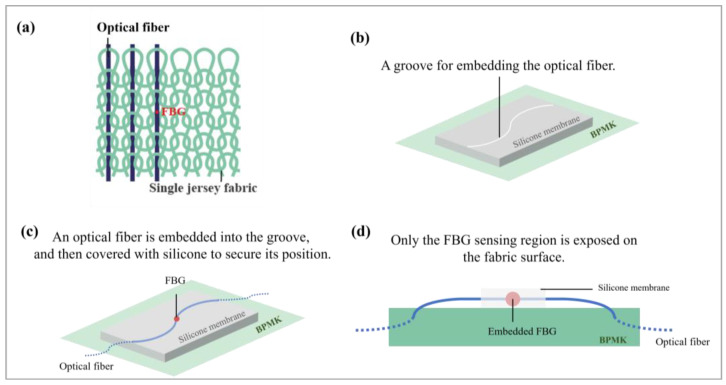
(**a**) Structure of optical fiber inlaid into single jersey fabric in warp direction, (**b**) silicone membrane with a groove, (**c**) FBG sensor embedded in silicone membrane, and (**d**) cross-section of BPMK.

**Figure 3 sensors-25-01284-f003:**
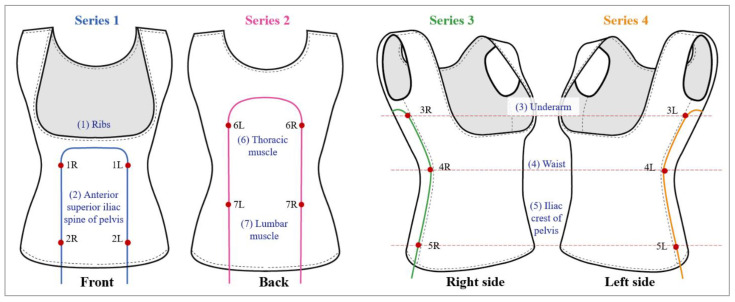
Placement of FBG sensors on BPMK.

**Figure 4 sensors-25-01284-f004:**
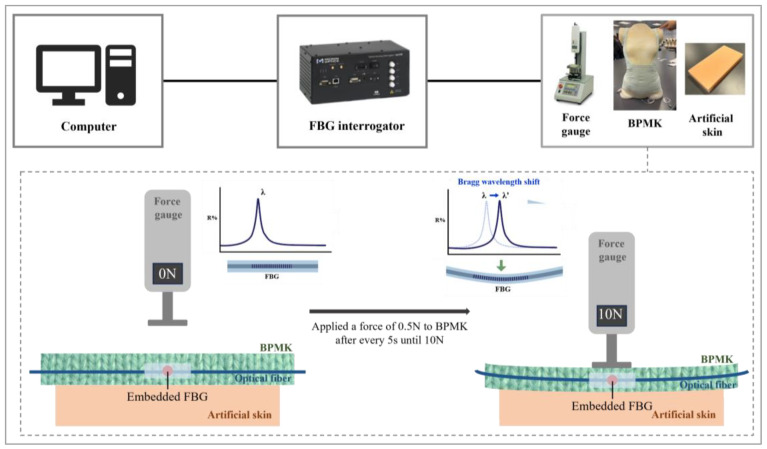
Equipment setup and schematic of transverse force applied to BPMK on artificial tissues.

**Figure 5 sensors-25-01284-f005:**
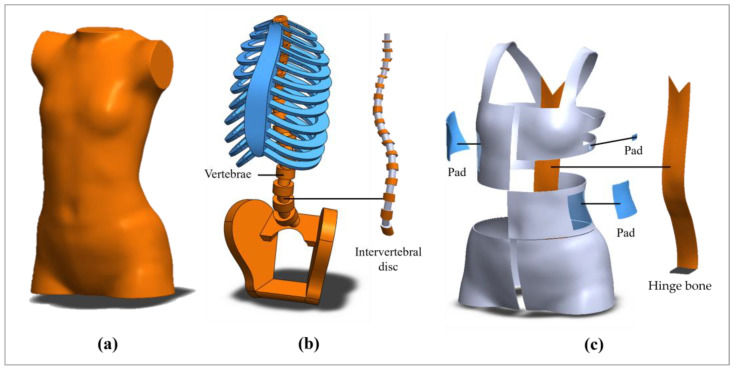
Models of: (**a**) torso, (**b**) skeletal structure, (**c**) textile materials and hinge bone of FIA.

**Figure 6 sensors-25-01284-f006:**
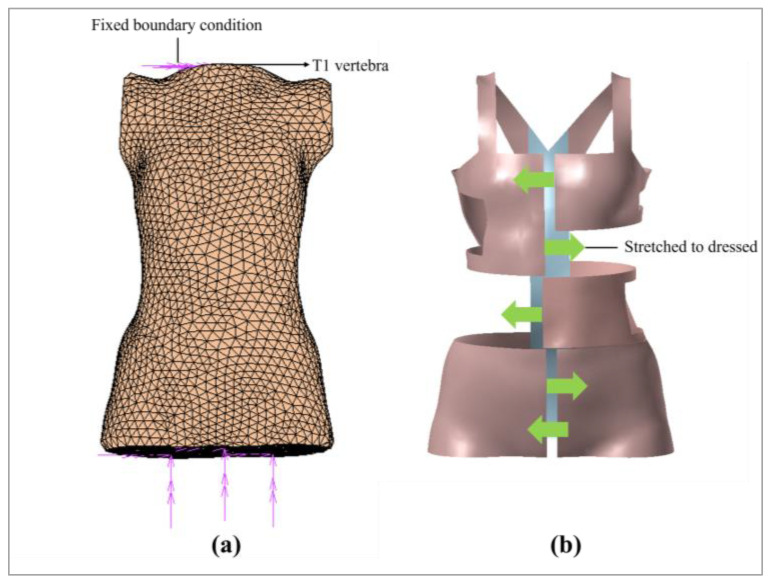
(**a**) Fixed boundary condition: all six degrees of freedom were set to zero, and (**b**) stretched boundary condition: the FIA was stretched to the dressed state.

**Figure 7 sensors-25-01284-f007:**
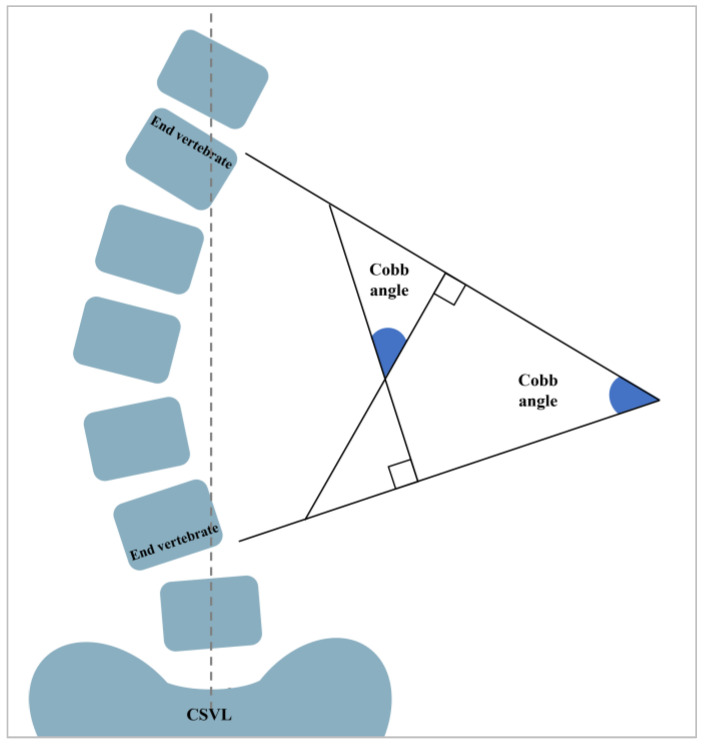
Schematic of Cobb angle measurements.

**Figure 8 sensors-25-01284-f008:**
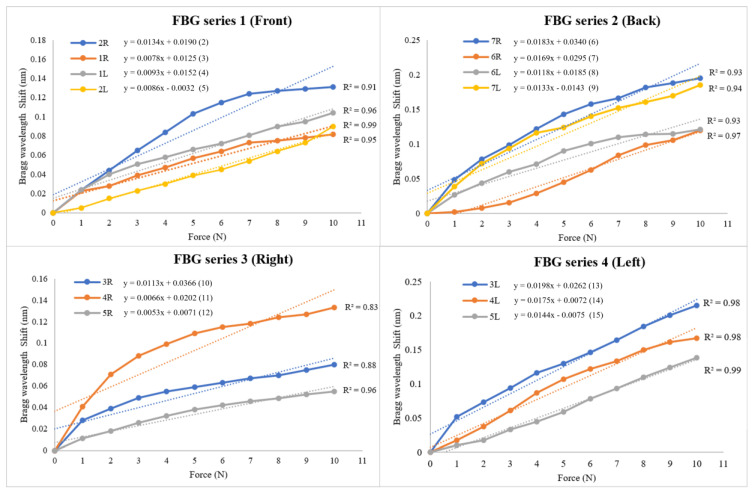
Linear regression between Bragg wavelength shift and force when applying force to different FBG Series 1 to 4.

**Figure 9 sensors-25-01284-f009:**
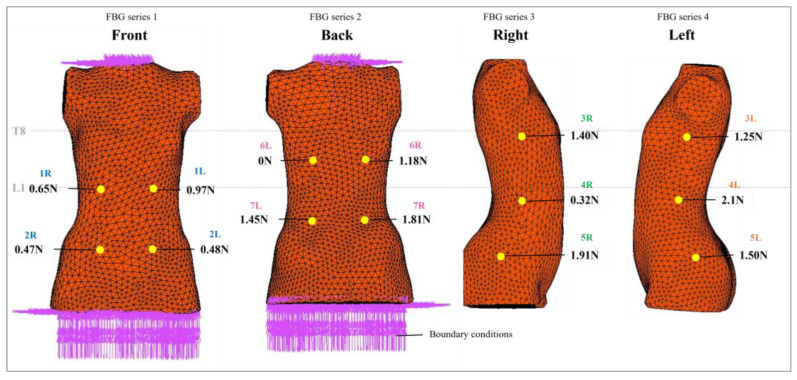
Force on interface between torso and FIA.

**Figure 10 sensors-25-01284-f010:**
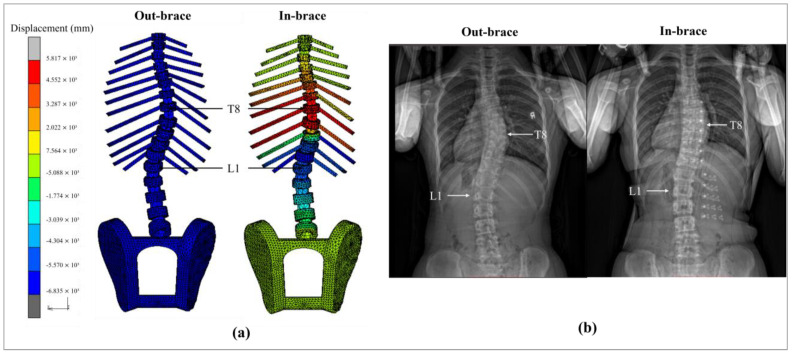
Displacement of skeletal model on: (**a**) FE model, and (**b**) X-ray images.

**Figure 11 sensors-25-01284-f011:**
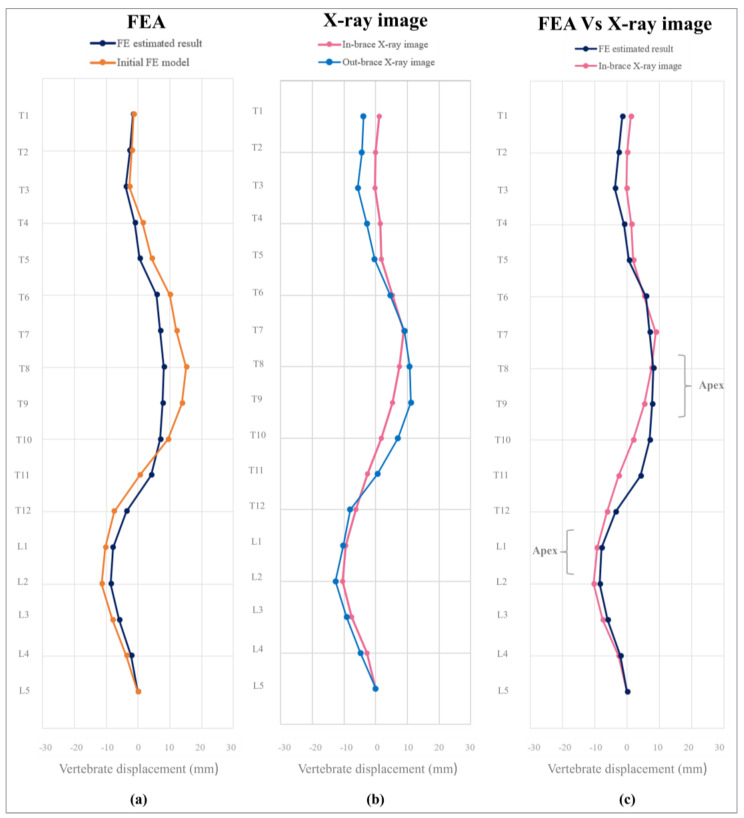
Comparison of spinal curves: (**a**) FEA, (**b**) X-ray images, and (**c**) their in-brace effects.

**Table 1 sensors-25-01284-t001:** Demographic data of subject before and after intervention [4].

	Before	After
Age	12	13
Body Mass Index (BMI)	20.2	/
Risser grade	2	2
Cobb angle at T5-T11 (°)	25.3	18.7
Cobb angle at T11-L3 (°)	21.8	18.8

**Table 2 sensors-25-01284-t002:** Specifications of FBG sensors.

FBG Series 1 (Front)		FBG Series 2 (Back)		FBG Series 3 (Right)		FBG Series 4 (Left)	
Location	λ (nm)	Location	λ (nm)	Location	λ (nm)	Location	λ (nm)
2R	1550.113	7R	1549.738	3R	1550.127	3L	1549.997
1R	1552.090	6R	1551.852	4R	1551.797	4L	1551.901
1L	1554.117	6L	1553.915	5R	1553.980	5L	1553.995
2L	1555.799	7L	1555.996				

**Table 3 sensors-25-01284-t003:** Mechanical properties of FE models.

	FE Model	Young’s Modulus (MPa)	Poisson’s Ratio
Torso	Body tissue	0.05	0.40
Skeletal-structure	Vertebrae, rib cage, pelvis	10,000	0.30
	Intervertebral disc	1.00	0.30
FIA	Textile material	0.59	0.40
	Hinge material	2070	0.30

**Table 4 sensors-25-01284-t004:** Force exerted by FIA.

FBG Series 1 (Front)		FBG Series 2 (Back)		FBG Series 3 (Right)		FBG Series 4 (Left)	
Location	Force (N)	Location	Force (N)	Location	Force (N)	Location	Force (N)
2R	0.471	7R	1.810	3R	1.400	3L	1.250
1R	0.648	6R	1.178	4R	0.324	4L	2.107
1L	0.965	6L	0.000	5R	1.905	5L	1.494
2L	0.481	7L	1.453				

**Table 5 sensors-25-01284-t005:** Comparison of Cobb angles from FEA and X-ray images.

	FEA		X-Ray Images	
	Before	After	Before	After
Cobb angle at T5-T11 (°)	24.9	18.1	25.3	18.7
Cobb angle at T11-L3 (°)	22.1	16.7	21.8	18.8
Apex	T8-9, L1	T8-9, L1	T8-9, L1	T8-9, L1

## Data Availability

The original contributions presented in this study are included in the article. Further inquiries can be directed to the corresponding author.

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
