# Peer review of "Enhancing Orthotic Treatment for Scoliosis: Development of Body Pressure Mapping Knitwear with Integrated FBG Sensors"

_sensors, 2025, doi:10.3390/s25051284_

Round 1
Reviewer 1 Report
Comments and Suggestions for Authors
Enhancing Orthotic Treatment for Scoliosis: Development of Body Pressure Mapping Knitwear with Integrated FBG Sensors
This article is devoted to the study of the effect of corrective corsets on the Cobb angle and spinal curvature. The study is based on the FE model developed in the Fok program.
The authors of the work are engaged in the design and creation of a knitted product - a corset with a pressure sensor system (BPMK). 14 FBG sensors located on the surface of the corset allow measuring both the general and local pressure exerted by the fixing elements.
To test the effectiveness of the developed corset, it was tested on a soft mannequin simulating a patient with adolescent idiopathic scoliosis (AIS). The results obtained will help to evaluate the effect of the corset on the correction of spinal deformity. The article is well structured, one there are a number of remarks.
- In the abstract, "FBG" should be spelled out in full at its first mention to ensure clarity for readers unfamiliar with the term.
- the main text, abbreviations such as "FE" (likely referring to "Finite Element") and "BMI" (Body Mass Index) should be explicitly defined at their first occurrence to enhance readability and prevent ambiguity.
-In Table 3, a period should be inserted in the line: "Skeletal structure Vertebrae, rib cage, pelvis 10,000" to maintain proper formatting and readability.
-In line 270, "the T1 vertebra" is mentioned. To improve clarity, it would be beneficial to include an indication or label in the corresponding figure to clearly show the location of the T1 vertebra.
-The quality of Figure 8 should be improved to ensure better visibility and readability of key details. Higher resolution or enhanced contrast might help in making the figure more informative.
-Consider adding numerical values of the obtained results to the figure annotations where applicable. This will provide readers with clearer, data-driven insights without requiring them to refer back to the text.
-In line 271, insert "the" before "bottom" to correct the grammatical structure.
- While the use of a validated soft AIS mannequin is a strength, it would be beneficial to provide more details on how its mechanical properties compare to actual human tissue, particularly regarding stiffness and deformation under load.
- Further testing on human subjects or a more detailed discussion on the feasibility of human trials would enhance the practical applicability of the findings.
- The prediction equations for FBG sensors should be explained more clearly, with additional validation against real-world clinical data, if possible.
Author Response
|
Comment 1: In the abstract, "FBG" should be spelled out in full at its first mention to ensure clarity for readers unfamiliar with the term.
Comment 3. In Table 3, a period should be inserted in the line: "Skeletal structure Vertebrae, rib cage, pelvis 10,000" to maintain proper formatting and readability.
Comment 5. The quality of Figure 8 should be improved to ensure better visibility and readability of key details. Higher resolution or enhanced contrast might help in making the figure more informative.
Comment 6. Consider adding numerical values of the obtained results to the figure annotations where applicable. This will provide readers with clearer, data-driven insights without requiring them to refer back to the text.
Comment 7. In line 271, insert "the" before "bottom" to correct the grammatical structure.
Comment 8. While the use of a validated soft AIS mannequin is a strength, it would be beneficial to provide more details on how its mechanical properties compare to actual human tissue, particularly regarding stiffness and deformation under load.
|
Comment 9. Further testing on human subjects or a more detailed discussion on the feasibility of human trials would enhance the practical applicability of the findings.
|
Response 9: Thank you for your insightful comments. We acknowledge that while our study demonstrates promising results using a validated soft AIS mannequin, conducting trials with actual human participants is essential.
We believe that human trials are feasible, and this paper represents an initial step toward that goal. However, improvements are necessary. We identified key limitations, such as the wearability of the device; for instance, the large interrogator can hinder the subject's movement during experiments. Additionally, the fragility of the silica optical fiber and the design of the undergarment need enhancement to reduce breakage risk. To address these challenges, we propose using a miniature interrogator, polymer optical fibers, and a design with front openings in the undergarment.
To highlight the importance of the feasibility of human trials, the limitations and proposed solutions are detailed in the 'Limitations of Experiments and Future Work' section. |
Comment 10. The prediction equations for FBG sensors should be explained more clearly, with additional validation against real-world clinical data, if possible.
|
Response 10: Thank you for your valuable feedback. Additional discussion has been added in Section 3.1 to further explain the prediction equations for the FBG sensors, and in Section 3.2 to compare the collected data with the clinical data. |

Reviewer 2 Report
Comments and Suggestions for Authors
(1) In the study, the authors developed a volumetric pressure mapping knitted fabric integrated with fourteen silicone embedded fiber Bragg grating sensors to monitor real-time and overall force changes during brace treatment. However, the abstract only presents qualitative results or conclusions, lacking clear explanations of quantitative results. For example, research results indicate that the measured force is highly consistent with the force obtained from clinical studies. What are the measurement results and clinical outcomes?
(2) The manuscript mentions that the methods used for continuous monitoring of applied pressure have limitations and may cause discomfort due to limited wearability. So, how has the comfort of the methods or solutions proposed in this study been overcome? And how is its comfort evaluated? What parameters are used to obtain the quantitative evaluation? Can similar comfort and accuracy evaluation results be obtained through experiments and clinical trials using dummies?
(3) In Table 3, we can see that almost all material authors in the model assumed it to be an elastic material. This is because in Figure 3, the author only provided the elastic modulus and Poisson's ratio of the relevant materials as two mechanical parameters. But in reality, all kinds of materials in nature are elastic-plastic. Is this assumption reasonable and accurate?
(4) What tools or methods were used in the research? Business software or self programming? If it is self transformation, the reviewer believes that specific relevant theories and solution methodologies need to be provided in the manuscript. If it is commercial software (such as ABAQUS or COMSOL), relevant information about the software or program needs to be provided.
(5) Figure 9 needs to provide relevant legends to display the specific physical meanings represented by different color blocks.
(6) From Figure 10, it can be seen that the deformation of bones is at the millimeter level. Such small results are easy and accurate to obtain for numerical simulations. However, for clinical practice, subtle measurement errors can lead to significant differences. How was this challenge overcome in the research?
Author Response
|
Comment 1: In the study, the authors developed a volumetric pressure mapping knitted fabric integrated with fourteen silicone embedded fiber Bragg grating sensors to monitor real-time and overall force changes during brace treatment. However, the abstract only presents qualitative results or conclusions, lacking clear explanations of quantitative results. For example, research results indicate that the measured force is highly consistent with the force obtained from clinical studies. What are the measurement results and clinical outcomes?
Comment 3. In Table 3, we can see that almost all material authors in the model assumed it to be an elastic material. This is because in Figure 3, the author only provided the elastic modulus and Poisson's ratio of the relevant materials as two mechanical parameters. But in reality, all kinds of materials in nature are elastic-plastic. Is this assumption reasonable and accurate?
Comment 5. Figure 9 needs to provide relevant legends to display the specific physical meanings represented by different color blocks.
|

Round 2
Reviewer 2 Report
Comments and Suggestions for Authors
This paper can be accepted for publication now.